# Potential Therapies Targeting the Metabolic Reprogramming of Diabetes-Associated Breast Cancer

**DOI:** 10.3390/jpm13010157

**Published:** 2023-01-14

**Authors:** Hang Chee Erin Shum, Ke Wu, Jaydutt Vadgama, Yong Wu

**Affiliations:** 1Li Ka Shing Faculty of Medicine, The University of Hong Kong, Hong Kong, China; 2David Geffen UCLA School of Medicine and UCLA Jonsson Comprehensive Cancer Center, Division of Cancer Research and Training, Department of Internal Medicine, Charles R. Drew University of Medicine and Science, 1748 E. 118th Street, Los Angeles, CA 90095, USA

**Keywords:** diabetes, breast cancer, cancer metabolism, metabolic reprogramming, hyperglycemia

## Abstract

In recent years, diabetes-associated breast cancer has become a significant clinical challenge. Diabetes is not only a risk factor for breast cancer but also worsens its prognosis. Patients with diabetes usually show hyperglycemia and hyperinsulinemia, which are accompanied by different glucose, protein, and lipid metabolism disorders. Metabolic abnormalities observed in diabetes can induce the occurrence and development of breast cancer. The changes in substrate availability and hormone environment not only create a favorable metabolic environment for tumorigenesis but also induce metabolic reprogramming events required for breast cancer cell transformation. Metabolic reprogramming is the basis for the development, swift proliferation, and survival of cancer cells. Metabolism must also be reprogrammed to support the energy requirements of the biosynthetic processes in cancer cells. In addition, metabolic reprogramming is essential to enable cancer cells to overcome apoptosis signals and promote invasion and metastasis. This review aims to describe the major metabolic changes in diabetes and outline how cancer cells can use cellular metabolic changes to drive abnormal growth and proliferation. We will specifically examine the mechanism of metabolic reprogramming by which diabetes may promote the development of breast cancer, focusing on the role of glucose metabolism, amino acid metabolism, and lipid metabolism in this process and potential therapeutic targets. Although diabetes-associated breast cancer has always been a common health problem, research focused on finding treatments suitable for the specific needs of patients with concurrent conditions is still limited. Most studies are still currently in the pre-clinical stage and mainly focus on reprogramming the glucose metabolism. More research targeting the amino acid and lipid metabolism is needed.

## 1. Background

Diabetes mellitus (DM) has always been one of the most common health conditions. In 2022, more than 11% of the US population was diagnosed with DM [1]. Meanwhile, breast cancer (BC) has the highest incidence among women’s cancers [2]. Epidemiology studies have shown that DM increased BC risk in women by up to 20% [3] and raised its mortality rate significantly [4]. There are two main types of DM: Type 1 and Type 2. T1DM, also known as insulin-dependent DM, is caused by a lack of insulin production which hinders glucose uptake in the body. On the other hand, T2DM is described as insulin independent. It usually happens later in life when cells develop insulin resistance due to a sedentary lifestyle and poor dietary habits. Between 90% and 95% of DM belongs to T2DM.

The insulin resistance in DM causes hyperinsulinemia and hyperglycemia, further disrupting the body’s regular metabolism. On the other hand, metabolic reprogramming occurs in cancer cells to enhance nutrient absorption, hence supporting their rapid proliferation rate. Recent therapeutic target research has focused on the similarities between the metabolic reprogramming of DM and BC, with the three main focuses being glucose, amino acid, and lipid metabolism [5]. Given the high prevalence of both diseases and the evident metabolic linkage between them, it is essential to discover a breakthrough treatment for this patient group. This review summarizes the mechanism of metabolic reprogramming that DM promotes the occurrence/development of BC and the corresponding therapeutic metabolic reprogramming strategy (MRS) for DM-associated BC.

## 2. MRS Targeting Glucose Metabolism

It is shown that hyperglycemia and hyperinsulinemia in DM promote BC progression [5]. Hyperglycemia supports BC cell growth by providing sufficient glucose for aerobic glycolysis, known as the “Warburg effect”. Under aerobic glycolysis, a large amount of lactate is produced to generate enough ATP to support rapid cancer cell proliferation [6]. Hyperinsulinemia in DM also promotes BC cell growth. Besides inducing glucose uptake, insulin plays a vital role in activating mTOR via the PI3K/AKT pathway. Activating mTORC1/2 increases mRNA translation, cellular growth, and cell proliferation and enhances cell survival [7]. Within the pathway, Akt and phospholipase Cγ play a key role in BC patient with diabetic conditions [8]. Many components of the metabolic pathway, such as transporters and kinases, have been found to have significant therapeutic potential as MRS targets for DM-associated BC. The possible target in glucose metabolism pathway for metabolic reprogramming of breast cancer is summarized in Figure 1.

### 2.1. Metformin

Metformin is best known as a diabetic drug, but it has also been proved useful as a treatment in BC. It has two main mechanisms to carry out its anti-diabetic and anti-tumorigenic effect: the AMPK-dependent and -independent pathway. In the AMPK-dependent pathway, AMPK increases glucose uptake and reduces gluconeogenesis, thus improving glycemic control. Moreover, it inhibits mTORC1, which induces BC cell proliferation. For the AMPK-independent pathway, metformin regulates oncogenes and tumor suppressor genes. Anti-tumorigenic effects targeting the reactive oxygen species, NF-kB, and cell cycle regulatory proteins are also carried out [9]. However, despite the encouraging results of metformin on BC from preclinical research, several clinical trials on BC patients without DM show no significant improvement in patient survival [10,11]. Further clinical trials are required on DM-associated BC patients to better understand its therapeutic potential.

Recent studies have shown that when metformin is used in combination with other therapeutic drugs, its effectiveness will be improved. For example, glucagon-like peptide-1 receptor agonist exendin-4 (Ex-4), another anti-diabetic drug, has been proven to be effective for BC when used in combination with metformin [12] by inhibiting NF-κB [13] and modulating different RNA gene expression [12].

MCT4 inhibitor is another potential drug candidate that can be used in combination with metformin. As lactate is produced from aerobic glycolysis in cancer cells, MCT4 maintains intracellular pH balance by exporting lactate out of cells. A novel MRS uses metformin and NF-κB inhibitor to further enhance the rate of aerobic glycolysis and increase the amount of lactate produced. With the addition of an MCT4 inhibitor, lactate accumulation decreases intracellular pH and achieves cytotoxicity by intracellular acidification [14]. Another study has shown that BC chemotherapies, such as paclitaxel and doxorubicin, have a lower efficacy in diabetic patients under metformin treatment as a large amount of lactate produced is pumped out of cells by MCT4, hence inducing extracellular acidosis, which inhibits doxorubicin’s uptake. MCT4 inhibitors can potentially improve chemotherapy response in DM-associated BC patients by regulating the extracellular pH [15].

### 2.2. GLUT Inhibitor

Glucose is one of the most important energy sources of BC cells, especially under the hyperglycemic condition in DM patients. Therefore, regulating glucose transporters is another research direction for DM-associated BC. Fourteen glucose transporters have currently been discovered [16]. This review will focus on GLUT1-4 and GLUT12, commonly expressed in BC, showing therapeutic potential in BC patients with concurrent DM.

GLUT1 is well known to be upregulated in BC cell lines [17] and plays a significant role in glucose uptake in BC tissues. The oncogenic factors in DM, such as insulin, glucose, INF-γ, and oxidative stress, are found to be mediated by GLUT1 to produce its effect [18]. A recent study has confirmed the effectiveness of GLUT1 inhibitors in hindering the growth of triple-negative breast cancer (TNBC), one of the deadliest subtypes of BC, by inhibiting glucose influx to undergo glycolysis [19]. Given the great potential of a GLUT1 inhibitor in BC, there are numerous ongoing studies about the inhibition of GLUT1, involving both synthetic agents, natural compounds and registered drugs such as metformin [20].

GLUT2, as a low-affinity glucose transporter, works well in high glucose concentrations [21]. It is also found to be present in some BC cell lines. Natural compounds, including phloretin and cytochalasin B, are effective as GLUT2 inhibitors to reduce glucose uptake in BC cell lines such as MDA-MB-231 [22] and MCF-7 [23].

GLUT3 is also a transporter in numerous cancer cells, especially under hypoxic conditions [17]. New findings have pointed out that downregulation of it can inhibit glycolysis and proliferation of BC [24]. It is a poor prognostic factor of BC [25].

GLUT4 is an insulin-dependent transporter [26]. It translocates to the membrane only in the presence of insulin. GLUT4 inhibition induces metabolic reprogramming, shifts glycolysis to oxidative phosphorylation, and lowers BC’s proliferation rate under hypoxic conditions [27]. However, studies have shown that the downregulation of GLUT4 worsens DM progression as it hinders glucose uptake and causes peripheral insulin resistance and poorer glycemic control [28]. Recent studies have shown that, different from BC, it is important to increase the expression of GLUT4 via pharmaceutical and dietary means to improve the diabetic condition [29]. Therefore, despite the success of GLUT4 inhibition therapy in BC, GLUT4 may not be a suitable drug target for BC patients with concurrent DM.

GLUT12, an insulin and glucose-sensitive transporter, was first found in BC cell lines [30]. Lowering the expression of GLUT12 has been proven to inhibit TNBC cell proliferation by decreasing glucose uptake and inhibiting aerobic glycolysis [31]. More importantly, GLUT12 plays an essential role in BC progression under hyperglycemic conditions by detecting high glucose environments and assisting in the migration of MCF-7 cells [32]. However, GLUT12, the same as GLUT4, is an insulin-sensitive glucose transporter. A high level of GLUT12 is essential to maintain insulin sensitivity in diabetic patients [33], contraindicating the fact that inhibiting GLUT12 helps treat BC cells. Therefore, GLUT12 may not be a suitable target for DM-associated BC.

### 2.3. SGLT Inhibitor

Sodium-glucose cotransporter-2 (SGLT2) is a key transporter located in the proximal convoluted tubules of the kidneys [34]. It is responsible for glucose reabsorption and plays an essential role in the glycemic control of our body [34]. As its name suggests, the transporter exports glucose from the cell to circulation with the help of sodium ions to create an electrochemical gradient [35]. Given its importance in glycemic control, several SGLT2 inhibitors are already approved by FDA and used in DM patients, including canagliflozin, ipragliflozin, empagliflozin, dapagliflozin, and ertugliflozin. They work by inhibiting glucose reabsorption in the kidney and increasing renal glucose excretion [36].

SGLT2 inhibitors are also found to be effective in treating BC via various mechanisms. Although SGLT2s are typically expressed in renal cells, several studies have discovered their presence in BC cells [37,38]. Ipragliflozin can inhibit glucose and sodium influx into cells, hyperpolarize cancer cells’ membranes, and hinder cancer cell growth [37]. Other SGLT2 inhibitors, Canagliflozin and Dapagliflozin, can inhibit BC proliferation by inducing nutrient deficiency and cell cycle arrest [38]. Dapagliflozin is also an effective agent in ameliorating hyperinsulinemia, which causes BC progression in DM patients [39]. However, the inhibitors’ effect is limited to BC with a specific mutation, namely Pten-driven EMT6 tumors and HRAS-driven Ac711 tumors [40]. Several clinical trials are ongoing to determine the safety and efficacy of SGLT2 inhibitors when used in combination with other BC chemotherapy (Table 1). Although more preclinical and clinical research is required, SGLT2 inhibitors have considerable potential as MRS for DM-associated BC.

Besides the SGLT2 inhibitor, we should not neglect the potential role of SGLT1 inhibitors in the therapy for DM-associated BC. SGLT1 mainly facilitates glucose transport in the small intestine and kidney [41]. Compared to the SGLT2 inhibitor, the SGLT1 inhibitor is a less popular study topic due to its minor contribution to glycemic control [34]. However, the SGLT1 inhibitor is a beneficial agent in enhancing the SGLT2 inhibitors’ effect in glycemic control of diabetic patients [41]. There are several dual inhibitors of SGLT1 and SGLT2, including LX4211 and Sotagliflozin. LX4211 shows satisfactory results in improving glycemic control in phase 1 clinical trials [42]. In phase 3 studies of Sotagliflozin, although it has been proven to be a safe drug, an insignificant anti-diabetic effect is shown [43]. When it comes to BC, overexpression of SGLT1 is found in tamoxifen-resistant ER-positive BC. The transporter increases glycolytic flux and lactate production via aerobic glycolysis which induces a tumor-associated macrophage. The macrophage then promotes cell growth via EGFR/PI3K/Akt signaling, releasing immunosuppressive factors [44]. In line with the previous study, there is other evidence suggesting that high expression of SGLT1 correlates to a high growth rate of TNBC [45]. Knocking down SGLT1 is proven effective in inhibiting BC growth, including subtypes such as TNBC [45] and HER2+ BC [46]. SGLT1 is a potential therapeutic target for DM-associated BC. However, discovery of more selective SGLT1 inhibitors against BC cells is required. More in-depth studies are necessary to determine the safety and efficacy of using SGLT1 and SGLT2 inhibitors in combination.

### 2.4. MCT Inhibitor

Monocarboxylate transporter (MCT) 1 & 4 are one of the leading research directions of BC therapy. Due to the Warburg effect, cancer cells shift their metabolism from oxidative phosphorylation to aerobic glycolysis, producing lactate for energy production. Under the hyperglycemic condition of DM, MCT plays a crucial role in maintaining the balance of lactate and the pH of cells [47]. Both MCT1 & 4 are upregulated in BC [48,49].

MCT1 is a bidirectional lactate transporter. A few MCT1 inhibitors (MCT1i) have already undergone phase I/II clinical trials for solid tumors. As a sole agent, MCT1 is found to have no direct toxicity toward BC cells. The blockage of the lactate import of MCT1 is more effective under glucose deprivation, as cancer cells will switch back to glucose for energy production when there is a lack of lactate [50]. It is also shown that MCT1i’s effect may be compromised by the upregulation of MCT4, a lactate exporter. Therefore, combination therapy may be required to improve MCT1i’s efficacy [51]. A recent study suggests that inhibiting lactate import by MCT1i while depleting BACH1 proteins, which increases lactate catabolism, is effective as a TNBC therapy. As DM-associated BC cells rely heavily on aerobic glycolysis and lactate metabolism, the combination therapy forces cells to reprogram their metabolism to oxidative phosphorylation, which is less favorable for the rapid proliferation rate [52].

The MCT4 inhibitor (MCT4i) is also a popular candidate for DM-associated BC. As mentioned above, upregulation of MCT4 compromises the efficacy of MCT1i; therefore, a combination of MCT1i and MCT4i may be required [50]. As a lactate exporter, it is essential to note that MCT4 regulates intracellular and extracellular pH. We demonstrated that under the hyperglycemic condition, it can be used in combination with metformin and NF-κB inhibitors to achieve intracellular acidification [14]. It can also prevent extracellular acidification and inhibit the expression of acidity-sensitive immune checkpoint protein, hence acting as immunotherapy [53]. Although MCT4i has excellent therapeutic potential, some studies have shown that inhibition of MCT4i will increase the risk of endothelial injury and cardiovascular complications [54]. MCT4 inhibition may also induce resistance toward traditional BC chemotherapy such as tamoxifen [55]. More research is required to ensure its safety and efficacy in DM-associated BC.

### 2.5. Insulin Growth Factor Receptor (IGF1R) Antagonist and Insulin Receptor (IR) Antagonist

Insulin-like growth factor (IGF) and insulin are peptide hormones that regulate our growth and metabolism. Subunits of their respective receptors, IGF-1R, IR-A, and IR-B, form heterodimeric hybrid receptors. With their similar amino acid sequence, they are known to bind to each other’s receptors [56]. Besides glycemic controls, insulin could cause tumorigenesis and trigger the downward cascade involving the mTOR signaling pathway [57]. Therefore, insulin treatment in DM has always been controversial due to its potential risk of inducing cancer [58,59]. In more recent research, we found that the cysteine-rich 61 (Cyr61) elevation plays a key role in tumorigenesis. Once IGF-1 binds to IGF1R, it triggers the PI3K/AKT and MAPK signaling pathway, inducing the production of Cyr61. As IGF-1 is an essential growth factor, there are concerns that inhibiting IGF1R may affect other organs. It is suggested that targeting specific tumorigenic molecules along the IGF1 signaling pathway, such as Cyr61, will be a safer and more effective potential therapy [60]. Inhibition of Cyr61 signaling is proved to be effective in hindering BC growth [61]. An anti-Cyr61 antibody, 093G9, is proved to be an effective approach to inhibiting Cyr61 in vivo [62].

In DM-associated BC, hyperinsulinemia is one of the key factors which promotes cell proliferation and BC growth. IGF1R and IR are well-anticipated potential therapies for DM-associated BC. However, regulating IGF1R shows disappointing results in clinical trials. According to a review, most treatments show no improvement compared to the standard of care. Some hypothesized that an increase in insulin receptor expression compromises the drug’s efficiency [63]. With an increasing number of BC MRS targeting IR, e.g., anti-idiotypic antibodies [64], combined therapy of IGF1R and IR antagonists may be able to optimize the therapeutic effect [65]. New findings indicated that combinations of IGF1R/IR with androgen receptor antagonist or anti-PD-L1, are effective in hindering migration and progression of TNBC cells [66,67]. However, IGF1R and IR therapies can cause worrying side effects. It is shown that low levels of IGF in the blood could lead to metabolic syndrome, insulin resistance, and glucose intolerance. Like insulin, IGF can also promote glucose uptake in many cell types. Antagonizing its receptor may further worsen the metabolic disturbance of DM patients [68]. IR also produces worrying side effects for DM patients. When IR antagonist S961 is used as a sole agent to treat BC, it results in hyperinsulinemia, hyperglycemia, and increased BC tumor size, possibly due to elevated amounts of insulin targeting the IGF1R that is not inhibited [65]. It is still unclear if the antagonist of IGF1R and IR is suitable for DM-associated BC. More research is required to investigate the potential side effects of IGF1R and IR antagonists on BC patients with DM.

## 3. MRS under Amino Acid Metabolism

Besides glucose, amino acids also play an essential role in the metabolism of DM and BC. There are 20 amino acids (AA) in total which can be divided into two groups: essential and non-essential AA. Essential AAs cannot be synthesized in our body and must be retrieved from the diet, and vice versa for non-essential AAs [69]. Each AA has their specific role in DM-associated BC. Certain AA can provide additional energy to support the rapid cancer cells proliferation. Some of them can also trigger cell signaling pathways which cause malignant cell proliferation. This review will focus on several AAs relevant to DM-associated BC therapies [70]. Table 2 summarizes the clinical trials of amino acid pathway treatment.

### 3.1. Leucine

Leucine is an essential AA that is involved in the pathogenesis of T2DM. Leucine can be obtained from a protein diet and act as an activator of mTORC1. Excessive protein in the diet for an extended period may lead to insulin resistance and T2DM due to early β-cell apoptosis [71]. P70 S6 kinase 1 (S6K1) is activated correspondingly, leading to insulin secretion. The activation of mTORC1 also increases cell proliferation, growth, and tumorigenesis by various signaling molecules such as the eIF4E-binding protein 1, suggesting leucine as a linkage between DM and BC [70].

L-type amino acid transporter (LAT1) inhibitor is a popular potential therapy for BC. LAT1 is a cancer prognostic marker in which a higher level corresponds to poorer survival [72]. Some also suggest that leucine can act as a nutrient source for energy production in cancer cells, augmenting the supply of glucose. Therefore, the increased expression of the transporter can explain the resistance of ER+ BC to chemotherapy [73]. A novel selective LAT1 inhibitor, JPH203 (Figure 2Ⓕ), effectively inhibits cell proliferation in estrogen-deprived BC [74]. JPH203 can also increase the sensitivity of BC cells to radiotherapy by inhibiting leucine uptake [75]. An approved drug ASCT2 inhibitor, Benzylserine, which manifests LAT1 inhibiting ability, also inhibits BC growth, further proving the therapeutic potential of inhibiting LAT1 [76]. However, there are still a lot of uncertainties regarding the use of LAT1 inhibitors in DM-associated BC. Nevertheless, some have proved that inhibiting leucine uptake and lowering its concentration in cells can prevent the progression of insulin resistance and DM [77]. Most agree that inducing leucine uptake benefits DM patients as it compensates for their β cell dysfunction and insulin resistance [78,79]. Leucine-rich diet is one way to increase the supply of leucine [80]. Further studies must be carried out to confirm the role of leucine in DM-associated BC.

### 3.2. Methionine

Methionine is an amino acid that plays a crucial role in protein synthesis, transmethylation reaction, and formation of metabolic molecules [81]. An increased level of methionine is often seen in DM. One of its primary intermediates, S-adenosylmethionine, is said to cause DNA methylation and DM progression [82]. Methionine also induces tumorigenesis by several mechanisms, via the synthesis of glutathione, formation of polyamine, and donation of a methyl group which promotes DNA methylation [83].

With the apparent relevance of methionine in both DM and BC, dietary methionine restriction (MR) could be an effective DM-associated BC strategy. One of the most recent MRS for TNBC involves MR. Dietary depletion of methionine causes BC cells to rely entirely on the thioredoxin antioxidant pathway instead of the pathway involving glutathione. Combined with a thioredoxin reductase agent, it exerts antitumor effects on BC by increasing oxidative cell stress [84]. A review of methionine restrictions also concluded that reducing the intake of methionine exerts an excellent anti-diabetic effect by increasing insulin sensitivity in different tissues, reducing inflammatory reactions and oxidative stress, etc. [82]. Other than dietary intervention, oral recombinant methioninase (o-rMETase) could also modulate methionine metabolism. Recent studies have confirmed that o-rMETase is effective in BC, especially in metastasized cancers [85], and reduces recurrence [86]. It can also prevent the onset of DM [87]. Given its benefits, MRS targeting methionine metabolism has tremendous therapeutic potential for DM-associated BC.

### 3.3. Cysteine

Cysteine is an amino acid closely related to methionine. In the methionine metabolism, methionine can be converted to homocysteine and then to cysteine after a series of reactions [81]. BC cells use cysteine to protect themselves from oxidative stress by producing glutathione (GSH). Given its elevated level in DM, it can be a potential therapy target for DM-associated BC [70].

Targeting the cysteine pathway, cysteine deprivation could cause programmed necrosis in cysteine-addicted basal-type TNBC. Cysteine depletion inhibits BC cells’ ability to produce glutathione, which protects cells against oxidative stress [88]. There are also MRS targeting BC cells which are independent of cysteine. HDAC8 inhibitors are found to sensitize the effect of cysteine deprivation mediated by PKCγ [89]. It is also proved that BC tumorigenesis can be regulated by SLC3A1 (Figure 2Ⓒ), a cysteine transporter, which reduces the ability of cancer cells to survive under oxidative stress [90].

Although elevated cysteine level is shown to do more harm than good in BC, it turns out that a high level of cysteine improves the diabetic condition. Therefore, cysteine may not be an ideal target for DM-associated BC therapies. Instead of depleting cysteine, cysteine supplementation is suggested in DM to increase glutathione synthesis. It prevents macrovascular complications by protecting cells against reactive oxidative species [91,92,93].

### 3.4. Glutamine and Glutamate

Glutamine and glutamate are non-essential amino acids that can be synthesized in our body. Glutamine can be converted to glutamate and vice versa. They are relevant in both DM and BC yet controversial as a treatment for concurrent DM and BC. Glutamine plays an important role in gluconeogenesis [94]. In DM, an elevated rate of gluconeogenesis from amino acid is seen under dysregulated glucagon stimulation. It is stated that increased gluconeogenesis could be a possible mechanism for causing hyperglycemia in DM. Glutaminase, which catalyzes the breakdown of glutamine, is shown to be beneficial to the glycemia control of DM patients [95]. However, glutamine supplementation is also suggested for DM as other studies proved that it improves fasting blood glucose and reduces cardiovascular risks [96,97].

Glutamine is suggested to be the most consumed AA for cell growth and proliferation [98]. Besides gluconeogenesis, glutamine can be converted to 𝛼-ketoglutarate, which incorporates into the TCA cycle to compensate for the insufficient energy production by aerobic glycolysis. Glutamate is also an essential intermediate for de novo amino acid synthesis [99]. In BC patients, reducing glutamine uptake is a means to inhibit TNBC and other BC cell growth. By inhibiting ASCT2/SLC1A5 (Figure 2Ⓑ), a transporter of glutamine, mTORC1 signaling, and cell cycle progression is halted, thus inhibiting cell proliferation [100,101]. With the disaccording results on the use of glutamine in DM, more research has to be conducted to determine whether it is a suitable pharmaceutical target for DM-associated BC patients.

### 3.5. Arginine

Arginine is a semi-essential amino acid. It has a substantial potential as a pharmaceutical target for DM-associated BC. It can be obtained through diet, synthesized endogenously, or produced from protein turnover. Arginine metabolism is essential as it plays a crucial role in protein production and the urea cycle, supporting cell growth [102]. By the action of nitric oxide synthase, arginine can also be converted to nitric oxide (NO), which plays an important role in tumorigenesis by numerous mechanisms, namely damaging DNA and promoting mutation on tumor suppressor gene, e.g., p53 and angiogenesis [103].

Recent studies have shown that BC cells take up arginine via the cationic amino acid transporter (CAT). CAT-1, CAT-2A, and CAT-2B have all been proven to be expressed in different BC cell lines [104]. With the identification of arginine transporter on BC cells, they are potential therapeutic targets to inhibit arginine uptake and BC cell growth. Besides its transporters, the enzymes breaking down arginine also have great potential as a therapy for BC. Novel agents, such as HuArgI (Co)-PEG5000 and arginine deaminase-PEG 20 (ADI-PEG 20) (Figure 2Ⓓ), have been discovered and proven to be effective when used against solid tumors such as breast [105], ovarian [106], and pancreatic cancers [107,108]. They break down arginine, causing disrupted metabolism and autophagy. Phase I clinical trial on ADI-PEG20 has already been carried out and demonstrated great results with its safety [105]. However, it is still unsure whether arginine is beneficial to BC. Different from what has been pointed out above, a study has found that arginine supplementation inhibits BC proliferation as it improves anti-tumor immunity via the activation of T cells [109]. With the combination of 5-fluorouracil and L-arginine supplementation, a synergistic effect against BC cells is found. They increase the programmed cell death rate and inhibit metastasis by altering the glycolytic metabolism [110].

Arginine supplementation is also a popular research topic in preventing diabetic complications. With NO being a well-known vasodilator and regulator of vascular resistance [111], it is hypothesized to be beneficial to microcirculation in DM patients and can reduce neuropathy incidence. However, the hypothesis was disproved in phase 4 clinical trial that it produces no significant effect on DM microcirculation [112]. An animal model has shown that L-arginine supplementation benefits renal function in DM patients and can enhance insulin sensitivity [113]. A recent study has found that higher arginine intake regularly may increase the risk of T2DM progression.

Given the controversial view of arginine’s role in DM-associated BC, it is important to note that NO acts differently in different concentrations. A low NO concentration promotes tumorigenesis, while a high level of it causes cytotoxicity and apoptosis of cancer cells [103]. More investigations on the dosage and possible contraindications are needed. It is also essential to clarify the role of arginine in BC patients with concurrent DM.

### 3.6. Serine

Serine is a non-essential AA that plays a key role in the metabolic reprogramming of cancer cells. Under the Warburg effect, cancer cells undergoing aerobic glycolysis require additional metabolism of AA for energy production to support their rapid proliferation rate. Along with glutamate, serine is a key player in biomass synthesis [98]. In one-carbon metabolism, serine, derived from glucose, is the donor of the one-carbon unit. The one-carbon unit is used to produce nucleotides, synthesize ATP and generate NADH/NADPH, which are all essential to support the rapid growth and proliferation of cancer cells [114]. The de novo serine synthesis pathway consists of enzymes that are an excellent target for MRS. Phosphoglycerate dehydrogenase (PHGDH) is the first rate-limiting enzyme in the pathway. It is proved that suppression of PHGDH inhibits BC growth, especially when targeting against the ER subtype and cells with high PHGDH levels. Even if we increase the supply of exogenous serine to cells, the effect of PHGDH suppression remains the same [115] (Figure 2Ⓐ). Suppression of other enzymes present in the de novo synthesis pathway, namely PSAT1 and PSPH, have also been shown to inhibit BC cell growth [115]. Novel PHGDH inhibitors have been screened out recently, which could play a huge role in future BC treatment. The newest compounds include CBR-5884 [116], C25 [117], and NCT-502 and NCT-503 [118]. All compounds have their pros and cons but it is concluded that most of them still show limited efficacy and selectivity [119]. Therefore, further investigations are required in the discovery of PHGDH inhibitors.

Besides BC, it is also essential to investigate the role of serine in DM patients with hyperglycemia. Although serine is derived from glucose, it is proven that a high amount of glucose will not increase the level of serine. A reduced amount of serine is found in DM patients [120]. Serine intake is found to improve insulin sensitivity and glucose tolerance. Serine supplementation is, therefore, a suggested therapy to improve their glucose tolerance [121]. As mentioned above, increasing serine extracellularly would not alter the effect of PHGDH suppression [115]. A study also indicated that the depletion of PHGDH can improve glycemic control in mice with obesity [122]. Therefore, combining serine supplementation and PHGDH suppression/inhibition may be a feasible therapy for DM-associated BC. However, contradictory findings indicate that chronic D-serine intake could adversely affect insulin secretion [123].

With the current evidence, targeting serine metabolism has exciting potential in treating DM-associated BC. However, more research is needed, specifically on discovering novel drug molecules and investigating their safety in DM patients.

### 3.7. Asparagine

As a non-essential amino acid, asparagine also plays a significant role in cell metabolism [124]. It could be a possible target in therapy for DM-associated BC. It acts as an AA exchange factor that facilitates the acquisition of other AA, namely serine and arginine, which play a huge role in cell growth and proliferation [125]. Unlike normal cells, tumor cells cannot synthesize L-asparagine and therefore rely on its external uptake for their growth. L-asparaginase works through consuming L-asparagine in plasma, which is necessary for leukemic cell survival. At present, three kinds of bacterial L-asparaginases are used for treatment, but they all have serious side effects related to high toxicity and immunogenicity (Figure 2Ⓔ). To lower asparagine levels in the body, it is suggested to inhibit asparagine synthetase (ASNS) or perform dietary asparagine restriction. Via the method mentioned above, lowering the asparagine level can inhibit the metastasis of BC [126]. By inhibiting the expression of asparagine synthetase, it is found to be effective in hindering BC cells’ growth [127]. However, there is yet a possible pharmaceutical molecule that can directly and selectively inhibit ASNS. Current research inhibits asparagine production by downregulating ASNS using small-interfering RNA (siRNA) [126,127]. After decades of limited progress, a recent study generated a high-resolution image of ASNS which rekindles the hope of discovering a selective ASNS inhibitor [128]. Asparagine is also closely related to glutamine metabolism. It is important to note that asparagine can compensate for glutamine depletion. When using glutamine restriction as an MRS, asparagine depletion or inhibition may be needed to fully disrupt the cancer cell metabolism [129]. However, as our review focuses on DM-associated BC, it is essential to note that asparagine improves insulin sensitivity [130]. A reduced level of asparagine has been proven to be associated with T2DM [131]. More data are needed regarding the status of asparagine in DM-associated BC patients and how BC therapy may affect concurrent diabetic conditions.

## 4. MRS Targeting the Lipid Metabolism

Lipids play an essential role in DM-associated BC. Under diabetic alteration of lipid metabolism, there is a ready supply of triglyceride and fatty acid (FA) [129]. FA is an important component for plasma membrane formation and plays a key role in cancer cell growth. It is a fuel for energy production and favors cell survival [5]. This review will investigate MRS targeting lipid metabolism in DM-associated BC.

### 4.1. De Novo FA Synthesis

FA can be produced inside the human body. The de novo FA synthesis is closely related to glucose metabolism. Pyruvate, a product of aerobic glycolysis, can be converted to acetyl-CoA and fatty acids under the influence of acetyl-CoA carboxylate and fatty acid synthase (FASN) [130]. FA is essential for cell membrane formation and is the building block of other lipids. It also plays a vital role in energy production for BC cells via β-oxidation.

Targeting the FA metabolism, the FASN inhibitor (FASNi) has excellent potential in DM-associated BC. An elevated level of FASN is seen in diabetic patients [131]. FASNi, platensimycin, exerts an excellent anti-diabetic effect as it concentrates in the liver. It alters metabolism by inhibiting lipogenesis and increasing glucose breakdown [132]. It is also a popular drug candidate for BC, especially against the HER2+ subtype [133]. HER2 overexpression induces FASN formation, while increasing FASN promotes HER2 signaling, forming a positive feedback [134]. It is also effective in resolving resistance of BC chemotherapies such as tamoxifen [135]. Its potential further increased with recent breakthroughs in discovering the mitochondrial priming mechanism of FASN. The classic FASNi includes Cerulenin, C75, and Orlistat, while novel agents such as TVB-3166 and TVB-2640 have been found recently [136]. Several specific to BC are going through phase 1 and 2 clinical trials [134]. Combination therapy with pro-apop6totic BH3 mimetic drugs further increases cancer cells’ sensitivity toward FASNi [137].

### 4.2. Exogenous FA Uptake

Besides synthesizing FA de novo, our body also obtains FA exogenously. CD36, also known as a fatty acid translocase, is responsible for the exogenous uptake and release of FA in numerous tissues such as adipocytes, skeletal muscle, and hepatic cells. It is known to be a biomarker of DM due to its elevated level. Several studies confirm its role in causing diabetic complications such as chronic kidney and atherosclerotic diseases [138,139]. Increased CD36 could induce insulin resistance and β cell dysfunction [140].

Expression of CD36 is also a poor prognostic marker in BC [141]. It provides FA, an alternative energy source, for cancer cells to grow and proliferate. CD36 is also the culprit in the resistance of HER2+ BC therapies [142]. Other than that, numerous findings indicated that CD36 depletion in cancer cells could inhibit their growth when combined with approaches that inhibit de novo FA synthesis. For example, using an SCD1 inhibitor with CD36 depletion prevents the uptake of exogenous monosaturated FA, a substance that reduces the SCD1 inhibitor’s effectiveness [143]. When using FASNi, there is a compensatory effect caused by CD36 upregulation, which desensitizes FASNi’s action to halt cancer cells from obtaining FA for cell proliferation [144]. With CD36 depletion, it can overcome resistance in different BC therapies.

Given its relevance in DM and BC, CD36 is a potential MRS target for DM-associated BC, primarily when combined therapy targets both the de novo FA synthesis and exogenous FA uptake. It benefits DM and treats BC in a nutrient-rich environment. Table 3 provides a comprehensive overview of the ongoing clinical trials that are currently being conducted to study the effects of FASN inhibitors and CD36 inhibitors.

## 5. Immune Dysregulation in Diabetes That Maybe Associated with Cancer Progression

In the presence of a tumor, anti-tumor immune cells mediate inflammatory responses to kill cancer cells after activating a metabolic switch. Nevertheless, tumors will develop strategies to avoid such damage. Cancer cells can change the metabolic environment of tumors through sequestering nutrients (such as glucose, tryptophan, and arginine) and producing toxic wastes (such as adenosine, lactic acid, and kynurenine). This tumor environment promotes the depletion of anti-tumor immune cells, the expansion of Tregs, and the expression of immune checkpoints [145,146,147]. The establishment of this immunosuppressive tumor environment reduces the therapeutic response of cancer patients to immunotherapy. The metabolic disorder under the condition of diabetes coordinates and aggravates this process [148]. In addition, diabetes mellitus itself can directly lead to immune dysregulation. For example, high glucose conditions in diabetes affect protein–oligosaccharide interactions via competitive inhibition [149]. Moreover, the increase in blood glucose level will form covalent sugar adducts with some proteins via nonenzymatic glycation. This can damage humoral immunity in various ways, for example, by changing the structure and function of immunoglobulins [150,151,152,153]. Previous studies have revealed that T cell function in patients with T2DM is impaired [154,155,156]. More and more evidence shows that abnormal metabolism in diabetes not only plays a key role in maintaining cancer signals of tumor occurrence and survival, but also has a wider significance in regulating anti-tumor immune response by releasing metabolites and influencing the expression of immune molecules, for instance, lactic acid, PGE2, and arginine [157]. Therefore, combination therapy between immunotherapy and metabolic intervention can be employed to better release the potential of anticancer therapy. Knowledge of immunometabolism allows novel therapeutic strategies to improve the anti-tumor immune response by targeting the metabolism of tumor and immune cells, so as to improve immunotherapy.

## 6. Conclusions

With the current trend of precision medicine, it is of utmost importance to design a personalized treatment plan for patients according to their specific conditions. Given that there are two concurrent conditions, DM and BC, it is essential to be aware of the contraindication of different therapy. With numerous similarities and clear linkages between DM and BC, it is surprising that there are still countless research gaps waiting for the scientific community to delve deeper into. This review shows different potential MRS targeting glucose, protein, and lipid metabolism, with glucose metabolism being the most popular research direction. Preclinical research of metformin and MCT1 & 4 inhibitors shows promising results when acting specifically on the metabolic reprogramming of diabetes-associated breast cancer. It is noteworthy that a lot of therapies may require a combination of drug molecules to achieve optimal results. Therefore, it is important to carry out further studies to confirm the best combination and be aware of any side effects from the combination of drugs. Further research on BC under diabetic conditions in vitro is needed, especially in the direction of protein and lipid metabolism. There are still many inconsistent findings about whether the therapies targeting amino acids such as leucine, glutamine, arginine, and serine are effective for diabetes-associated BC. Existing findings require more in vivo studies to test for their efficacy and clinical trials to ensure the safety of the therapies.

## Figures and Tables

**Figure 1 jpm-13-00157-f001:**
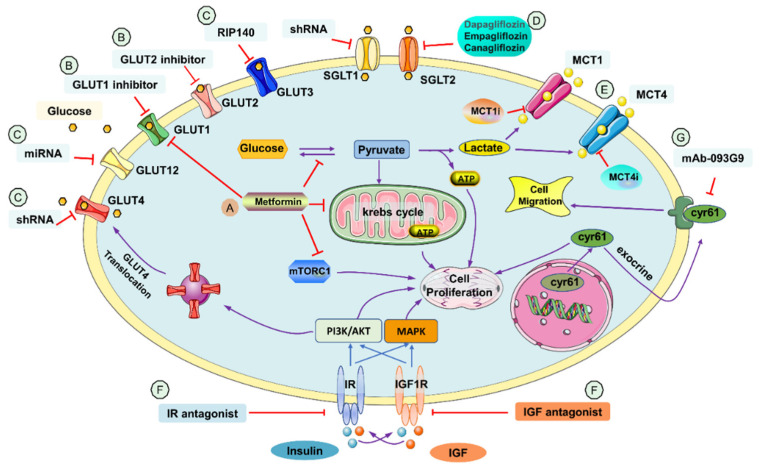
Potential targets of glucose metabolism pathway in metabolic reprogramming of breast cancer. Ⓐ Metformin’s mechanism of action in BC includes inhibiting GLUT1, inhibiting mTORC1, and shifting mitochondrial oxidative phosphorylation to aerobic glycolysis to increase the effect of MCT4i. Ⓑ GLUT1 and GLUT2 inhibitors inhibit the glucose influx to cancer cells. Ⓒ Downregulation of the transporter using transcriptional co-regulator RIP140, shRNA, and miRNA. There are no inhibitors of GLUT3, GLUT4 and GLUT12 suitable for BC yet. Ⓓ SGLT2 inhibitors inhibit the influx of glucose into BC cells. There is no selective SGLT1 inhibitor for BC yet but downregulation of the transporter by siRNA is found to be effective to inhibit BC growth. Ⓔ MCT1i inhibits the bidirectional transport of lactate. MCT4i inhibits lactate export, triggers intracellular acidification, and inhibits cell growth. Ⓕ IR and IGF-1R antagonists inhibit binding of insulin and IGF, thus inhibiting the mTOR signaling pathway which induces tumorigenesis. They also inhibit glucose uptake in cells. Ⓖ IGF1R triggers the PI3K/AKT and MAPK pathway which induces Cyr61 transcription and promotes BC growth. Antibody 093G9 is a potential therapy inhibiting Cyrs61.

**Figure 2 jpm-13-00157-f002:**
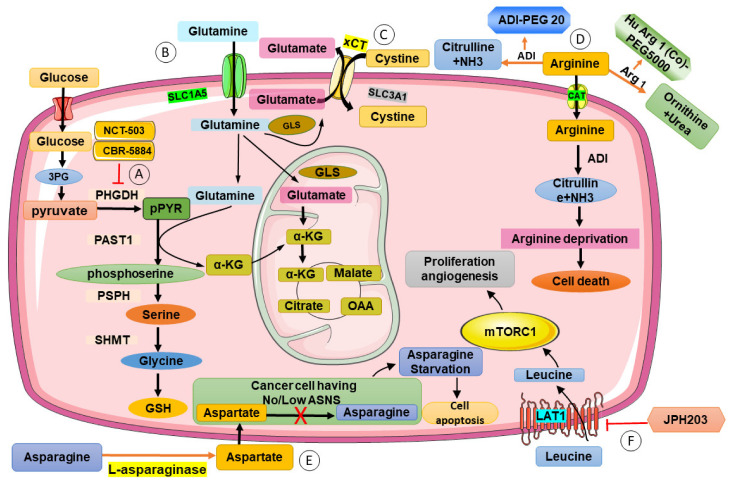
Possible MRS target in amino acid metabolism for DM-associated BC. Ⓐ In cancer cells, serine is used for energy production and biomass synthesis during aerobic glycolysis. Inhibiting enzymes that synthesize serine, such as phosphoglycerate dehydrogenase (PHGDH), may inhibit breast cancer growth. New inhibitors of PHGDH, such as CBR-5884 and NCT-503, have been developed and may have potential for use in breast cancer treatment. Ⓑ Glutamine and glutamate are non-essential amino acids that can be synthesized by the body and are involved in both diabetes and cancer. Glutamine can be converted to alpha-ketoglutarate, which is involved in the tricarboxylic acid cycle, while glutamate is necessary for the synthesis of new amino acids. In breast cancer, reducing the uptake of glutamine by inhibiting the ASCT2/SLC1A5 transporter has been shown to halt mTORC1 signaling and cell cycle progression, potentially inhibiting cancer cell growth. Ⓒ Cysteine is an amino acid that is involved in several important pathways in the body, including the metabolism of methionine and the production of glutathione. It has been proposed as a potential therapy target for BC, particularly for TNBC. Cysteine deprivation has been shown to cause programmed necrosis in BC cells that are dependent on cysteine, by inhibiting their ability to produce glutathione and protect against oxidative stress. Additionally, the cysteine transporter SLC3A1 has been found to regulate BC tumorigenesis by reducing the ability of cancer cells to survive under oxidative stress. Ⓓ Arginine is a semi-essential amino acid that has potential as a target for the treatment of diabetes-associated breast cancer. Arginine is taken up by cancer cells via cationic amino acid transporters, and enzymes that break down arginine, such as HuArgI (Co)-PEG5000 and arginine deaminase-PEG 20, have shown promise in treating solid tumors such as breast, ovarian, and pancreatic cancers. Ⓔ Asparagine is a non-essential amino acid that plays a role in cell metabolism and may be a potential target for the treatment of diabetes-associated breast cancer. Asparagine depletion or inhibition may be necessary to fully disrupt cancer cell metabolism when using glutamine restriction as a metabolic targeted therapy. L-asparaginase enzymes, which deplete plasma L-asparagine, have been used to treat acute lymphoblastic leukemia, but they can cause severe side effects due to their toxicity and immunogenicity. Ⓕ Leucine is an essential amino acid that may play a role in the development of type 2 diabetes and cancer. Leucine can activate the mTORC1 pathway, which is involved in cell proliferation, growth, and tumorigenesis. The L-type amino acid transporter (LAT1) is a cancer prognostic marker and a potential target for breast cancer treatment. LAT1 inhibitors such as JPH203 have been shown to inhibit cell proliferation and increase the sensitivity of breast cancer cells to radiation therapy. Figure abbreviation: PHGDH: phosphoglycerate dehydrogenase; 3PG:3-Phosphoglyceric acid; PAST1: Putative achaete scute target 1; PSPH: Phosphoserine phosphatase; SHMT: Serine hydroxymethyltransferase; GLS: glutaminase; α-KG: α-ketoglutarate; OAA: Oxaloacetic acid; LAT1: L-type amino acid transporter; ADI: Arginine deiminase; Arg1: Arginase 1.

**Table 1 jpm-13-00157-t001:** Ongoing clinical trials of SGLT2 targeting BC.

	Title	Status	Study Results	Conditions	Interventions	Phases	NCT Number
1	Alpelisib, Fulvestrant and Dapagliflozin for the Treatment of HR+, HER2-, PIK3CA Mutant Metastatic Breast Cancer	Recruiting	No Results Available	Metastatic BC HER2-negative BC	Dapagliflozin 10 Mg Tab	Phase 2	NCT05025735
2	A Phase 1b/2 Study of Serabelisib in Combination with Canagliflozin in Patients with Advanced Solid Tumors	Unknown status	No Results Available	BC Endometrial Cancer Lung Cancer Colorectal Cancer Head and Neck Cancer	Serabelisib Canagliflozin 300 mg	Phase 1 Phase 2	NCT04073680
3	Preventing High Blood Sugar in People Being Treated for Metastatic Breast Cancer	Recruiting	No Results Available	BC BC Stage IV Metastatic BC	Dietary Supplement: Ketogenic Diet Dietary Supplement: Low Carbohydrate Diet Drug: Alpelisib Drug: Fulvestrant Drug: Canagliflozin	Phase 2	NCT05090358
4	Study of Safety and Efficacy of Dapagliflozin + Metformin XR Versus Metformin XR in Participants With HR+, HER2-, Advanced Breast Cancer While on Treatment with Alpelisib and Fulvestrant	Recruiting	No Results Available	BC	Alpelisib Fulvestrant Metformin XR Dapagliflozin + metformin XR Dapagliflozin	Phase 2	NCT04899349

**Table 2 jpm-13-00157-t002:** Table of clinical trials related to DM-associated BC potential therapies targeting the amino acid pathway.

	Title	Status	Study Results	Conditions	Interventions	Phases	NCT Number
1	Arginine Metabolism in Pediatric Type 2 Diabetes	Recruiting	No Results Available	Type 2 Diabetes	Other: Stable isotope infusion, oral glucose ingestion, intravenous arginine bolus	Not Applicable	NCT05477134
2	SOAR-2: Intervening in Obesity Through Reduction of Dietary Branched Chain Amino Acids	Withdrawn	No Results Available	Obesity Diabetes	Control diet Dietary Supplement: Low branched-chain amino acids (BCAA) diet Dietary Supplement: Low protein diet	Early Phase 1	NCT04424537
3	Targeting Glutamine Metabolism to Prevent Diabetic Cardiovascular Complications	Recruiting	No Results Available	Glutamine Diabetic Cardiovascular Complications	Biological: Bio collection		NCT04353869
4	Effect of L-arginine on Microcirculation, Myogenesis and Angiogenesis Associated With Aging, Sarcopenia and Diabetes	Completed	No Results Available	Aging Sarcopenia Type 2 Diabetes Microcirculation	L-arginine Maltodextrin	Not Applicable	NCT04112875
5	The Effect of Pharmaceutical Grade L-glutamine (Endari) on Glycemic Control in Patients With Diabetes Mellitus Type II	Unknown status	No Results Available	Diabetes Mellitus, Type 2	Drug: L-glutamine Other: No L-glutamine	Phase 1	NCT03947879
6	Study of CB-839 (Telaglenastat) in Combination With Talazoparib in Patients With Solid Tumors	Terminated	Has Results	Solid Tumor Clear Cell Renal Cell Carcinoma TNBC Colorectal Cancer CRC|RCC| ccRCC	Drug: CB-839 Drug: Talazoparib	Phase 1 Phase 2	NCT03875313
7	Sulfasalazine in Decreasing Opioids Requirements in Breast Cancer Patients	Recruiting	No Results Available	Breast Cancer Chronic Pain Due to Malignancy (Finding)	Drug: Sulfasalazine Drug: Placebos	Phase 2	NCT03847311
8	ONC201 With a Methionine-Restricted Diet in Patients with Metastatic Triple Negative Breast Cancer	Terminated	Has Results	TNBC	Drug: Akt/ERK Inhibitor ONC201 Dietary Supplement: Methionine-Restricted Diet	Phase 2	NCT03733119
9	Study of Eryaspase in Combination with Chemotherapy Versus Chemotherapy Alone for the Treatment of TNBC (TRYbeCA-2)	Terminated	No Results Available	TNBC	Eryaspase (L-asparaginase encapsulated in red blood cells) Gemcitabine Carboplatin	Phase 2 Phase 3	NCT03674242
10	Methionine-Restricted Diet to Potentiate the Effects of Radiation Therapy	Suspended	No Results Available	Cancer Lung Cancer Prostate Cancer BC	Dietary Supplement: Methionine-restricted diet	Not Applicable	NCT03574194
11	Study of Arginine and Nitric Oxide in Patients with Diabetes	Completed	No Results Available	Ketosis Prone Diabetes	Dietary Supplement: Citrulline Dietary Supplement: Alanine	Not Applicable	NCT03566524
12	Effect of BKR-013 on Average Daily Glucose Levels	Completed	No Results Available	Type 2 DM	Other: BKR-013 or Placebo	Not Applicable	NCT03382015
13	Exercise Snacks and Glutamine to Improve Glucose Control in Adolescents with Type 1 Diabetes	Unknown status	No Results Available	DM, Type 1 Autoimmune Diseases DM Endocrine System Diseases Glucose Metabolism Disorders Immune System Diseases Metabolic Diseases	Drug: Glutamine vs. Placebo Other: Exercise	Not Applicable	NCT03199638
14	A Window of Opportunity Study of Methionine Deprivation in Triple Negative Breast Cancer	Terminated	No Results Available	BC TNBC	Dietary Supplement: hominex-2	Phase 2	NCT03186937
15	Study of CB-839 in Combination w/Paclitaxel in Participants of African Ancestry and Non-African Ancestry with Advanced Triple Negative Breast Cancer (TNBC)	Completed	Has Results	TNBC	Drug: Paclitaxel Drug: CB-839	Phase 2	NCT03057600
16	Arginase Inhibition and Microvascular Endothelial Function in Type 2 Diabetes	Completed	No Results Available	Type 2 DM	Other: N-hydroxy-nor-L-arginine	Phase 1 Phase 2	NCT02687152
17	Treatment of Type 2 Diabetes with Immunonutrients	Completed	No Results Available	DM, Type 2	Dietary Supplement: Arginine and fish oil	Not Applicable	NCT02462863
18	An Extension Protocol to Evaluate Dose Comparisons of Leucine-Metformin Combinations in Type 2 Diabetic Patients	Completed	Has Results	Type 2 DM	Low Metformin Mid Metformin High Metformin Metformin	Phase 2	NCT02435277
19	Novel Type 2 Diabetes Mellitus Preventive Therapies	Completed	No Results Available	Diabetes	Drug: Glutamine (Pharmacological doses) Behavioral: Lifestyle change	Phase 1	NCT02351323
20	Dose Comparisons of Leucine-Metformin Combinations on Blood Glucose Levels in Type 2 Diabetic Patients	Completed	Has Results	Type 2 DM	Low Metformin Metformin Mid Metformin High Metformin	Phase 2	NCT02151461
21	Study of the Glutaminase Inhibitor CB-839 in Solid Tumors	Completed	No Results Available	Solid Tumors TNBC Non-Small Cell Lung Cancer Renal Cell Carcinoma Mesothelioma Fumarate Hydratase (FH)-Deficient Tumors Succinate Dehydrogenase (SDH)-Deficient Gastrointestinal Stromal Tumors (GIST) Succinate Dehydrogenase (SDH)-Deficient Non-gastrointestinal Stromal Tumors Tumors Harboring Isocitrate Dehydrogenase-1 (IDH1) and IDH2 Mutations Tumors Harboring Amplifications in the cMyc Gene	Drug: CB-839 Drug: Pac-CB Drug: CBE Drug: CB-Erl Drug: CBD Drug: CB-Cabo	Phase 1	NCT02071862
22	Arginase Inhibition in Ischemia-reperfusion Injury	Completed	No Results Available	Coronary Artery Disease Type 2 DM	Drug: N-hydroxy-nor-arginine Drug: NaCl	Phase 1	NCT02009527
23	Ph 1 ADI-PEG 20 Plus Doxorubicin; Patients with HER2 Negative Metastatic Breast Cancer	Completed	No Results Available	HER2 Negative Metastatic BC	Drug: ADI-PEG 20	Phase 1	NCT01948843
24	L-Arginine, Vascular Response and Mechanisms	Completed	No Results Available	Hypertension DM	Dietary Supplement: L-Arginine Placebo Supplement	Phase 2	NCT01482247
25	Glutamine and Insulin Sensitivity in Type I Diabetes	Completed	Has Results	Type I DM	Dietary Supplement: Glutamine Placebo	Not Applicable	NCT01467063
26	Prevention of Type 2 Diabetes Mellitus by L-Arginine in Patients with Metabolic Syndrome	Completed	No Results Available	Metabolic Syndrome Impaired Glucose Tolerance |Insulin Resistance Endothelial Dysfunction	Drug: L-arginine Drug: Placebo	Phase 3	NCT00917449
27	Riluzole in Women with Stage I, Stage II, or Stage IIIA Breast Cancer	Withdrawn	No Results Available	BC	Drug: riluzole Genetic: polymorphism analysis Procedure: Axillary lymph node biopsy Digital image analysis Needle biopsy Sentinel lymph node biopsy Therapeutic conventional surgery	Phase 1	NCT00903214
28	Effect of Arginine on Microcirculation in Patients with Diabetes	Completed	No Results Available	Type 2 DM	Dietary Supplement: L-arginine Placebo Lactose powder	Phase 4	NCT00902616
29	Effects of Glutamine on GLP-1 and Insulin Secretion in Man	Completed	Has Results	Type 2 DM	Drug: Sitagliptin Drug: Placebo	Not Applicable	NCT00673894
30	N-Acetylcysteine and Arginine Administration in Diabetic Patients	Terminated	No Results Available	Type 2 DM Hypertension	Drug: Arginine Drug: Acetylcysteine Drug: Placebo	Phase 4	NCT00569465

**Table 3 jpm-13-00157-t003:** Ongoing clinical trials of FASNi and CD36.

	Title	Status	Study Results	Conditions	Interventions	Phases	URL
1	Nasturtium (Tropaeolum Majus L) Intake and Biochemical Parameters in Pre-diabetic Subjects in Bogota Colombia	Completed	No Results Available	Pre Diabetes	Dietary Supplement: Nasturtium (Tropaeolum majus)	Not Applicable	NCT05346978
2	FASN Inhibitor TVB-2640 and Trastuzumab in Combination with Paclitaxel or Endocrine Therapy for the Treatment of HER2 Positive Metastatic Breast Cancer	Recruiting	No Results Available	Advanced Breast Carcinoma HER2 Positive Breast Carcinoma Stage III Breast Cancer AJCC v7 Stage IIIA Breast Cancer AJCC v7 Stage IIIB Breast Cancer AJCC v7 Stage IIIC Breast Cancer AJCC v7 Stage IV Breast Cancer AJCC v6 and v7	Anastrozole Exemestane FASN Inhibitor TVB-2640 Fulvestrant Letrozole Paclitaxel Trastuzumab	Phase 2	NCT03179904
3	CD36 in Nutrient Delivery and Its Dysfunction	Completed	No Results Available	Insulin Resistance Endothelial Dysfunction	Sildenafil Citrate	Phase 1 Phase 2	NCT03012386
4	Evaluation of 3-V Bioscience-2640 to Reduce de Novo Lipogenesis in Subjects with Characteristics of Metabolic Syndrome	Completed	No Results Available	Metabolic Syndrome	3-V Bioscience-2640	Phase 1 Phase 2	NCT02948569
5	Inhibiting Fatty Acid Synthase to Improve Efficacy of Neoadjuvant Chemotherapy	Completed	Has Results	BC	Omeprazole	Phase 2	NCT02595372
6	Polymorphisms in CD36 and STAT3 Genes and Different Dietary Interventions Among Patients with Coronary Artery Disease	Unknown status	No Results Available	Coronary Artery Disease	Dietary Supplement: Olive oil Nuts Control diet	Not Applicable	NCT02202265
7	Metabolic and Cardiovascular Impact of CD36 Deficiency in African Americans	Completed	No Results Available	Obesity			NCT02126735
8	Proof of Principle Trial to Determine if Nutritional Supplement Conjugated Linoleic Acid (CLA) Can Modulate the Lipogenic Pathway in Breast Cancer Tissue	Completed	Has Results	BC	Conjugated Linoleic Acid (CLA)	Early Phase 1	NCT00908791

## Data Availability

No new data were created or analyzed in this study. Data sharing is not applicable to this article.

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
