# Peer review of "Potential Therapies Targeting the Metabolic Reprogramming of Diabetes-Associated Breast Cancer"

_jpm, 2023, doi:10.3390/jpm13010157_

Round 1
Reviewer 1 Report
Maior amendments
The abstract shoud involve some outcomes (the most significant and useful exaples, status/phase of studies), not only general knowledge.
Explain the role of the amino acids in DM and cancer (after head of the second paragraph, not only in the subheads) as well as summarize trials of amino acids patahway treatment, please.
Minor amendments
"1. MRS targeting glucose metabolism" - should be a separate chapter (edit letters style and line spacing, please).
Edit spacing, bold (f.e. delete bold in word "Sotagliflozin." (page 4)) and underlining (f.e. "more research has to be done to determine whether it is a suitable pharmaceutical target for DM-associated BC patients" (page 8)), please.
"MCT1i and MCT4i may be required. (Benyahia et al., 2021)." - change citing style, please (page 5).
Author Response
Responses to the Editor’s and Reviewers' Comments
Manuscript number: jpm-2098729
Title: Potential Therapies Targeting the Metabolic Reprogramming of Diabetes-Associated Breast Cancer
We would like to thank the editors and reviewers for your careful readings and for the thoughtful comments and constructive suggestions, which guided our revisions resulting in this improved paper. We are extremely encouraged to revise the manuscript according to reviewers' comments. Each comment has been carefully considered point by point and responded. We are confident that the new version of the manuscript will be greatly improved. We respond below in detail to each of the reviewer’s comments and also highlight the changes to the manuscript within the document by using Track Changes.
Reviewer: 1
Major amendments
- The abstract should involve some outcomes (the most significant and useful examples, status/phase of studies), not only general knowledge.
Response: We greatly appreciate the reviewer's insightful comments. As suggested by the reviewer, we have updated the abstract of our revised manuscript [marked] (Page 1).
- Explain the role of the amino acids in DM and cancer (after head of the second paragraph, not only in the subheads) as well as summarize trials of amino acids pathway treatment, please.
Response: We have added two sentences to explain the role of the amino acids in DM and cancer after head of the second paragraph of our revised manuscript [marked] (Page 7, the last paragraph). In addition, we summarized trials of amino acid pathway treatment by adding an updated Table 2 (Page 12) and Figure 2 (Page 11).
Minor amendments
- "1. MRS targeting glucose metabolism" - should be a separate chapter (edit letters style and line spacing, please).
Response: We have corrected it.
- Edit spacing, bold (f.e. delete bold in word "Sotagliflozin." (page 4)) and underlining (f.e. "more research has to be done to determine whether it is a suitable pharmaceutical target for DM-associ- ated BC patients" (page 8)), please.
Response: We have corrected them.
- "MCT1i and MCT4i may be required. (Benyahia et al., 2021)." - change citing style, please (page 5).
Response: We have corrected it.
Reviewer 2 Report
The review is aimed to establish a link between diabetes and breast cancer. Though the conceptual framework is sound, the review lacks presentation, content, and useful information, making it quite confusing for readers to grasp the review and its purpose. after the background, section 1.1-1.5 is pretty much redundant and meaningless as to why the authors described the inhibitors ( so many of them), without making any effort to link them to Cancer and oncogenesis. The entire section 2 is also an unnecessary description of amino acids without establishing what link it has with cancer. The review seems very confusing and hence not recommended for the journal in its present form unless completely rewritten. The authors made no effort to justify the altered metabolic environment in DM and how it contributes to cancer.The conclusion section lacks clarity.
Author Response
Responses to the Editor’s and Reviewers' Comments
Manuscript number: jpm-2098729
Title: Potential Therapies Targeting the Metabolic Reprogramming of Diabetes-Associated Breast Cancer
We would like to thank the editors and reviewers for your careful readings and for the thoughtful comments and constructive suggestions, which guided our revisions resulting in this improved paper. We are extremely encouraged to revise the manuscript according to reviewers' comments. Each comment has been carefully considered point by point and responded. We are confident that the new version of the manuscript will be greatly improved. We respond below in detail to each of the reviewer’s comments and also highlight the changes to the manuscript within the document by using Track Changes.
Reviewer: 2
- “ Section 1.1-1.5 is pretty much redundant and meaningless as to why the authors described the inhibitors (so many of them), without making any effort to link them to Cancer and oncogenesis.“
Response: We greatly appreciate the reviewer's comments. The focus of this paper, as indicated by its title, 'Potential Therapies Targeting the Metabolic Reprogramming of Diabetes-Associated Breast Cancer', is on potential targets for interventions that target the metabolic reprogramming mechanisms involved in the treatment of diabetes-associated breast cancer, rather than attempting to elucidate the mechanisms by which diabetes causes breast cancer. Therefore, the paper focuses on drug targets for interventions that target tumor metabolism in preclinical or clinical studies that have shown promise in the treatment of the diabetes-associated breast cancer population. Most of the drug targets involving metabolic reprogramming that have been identified to date are inhibitors of corresponding metabolic pathways, which inhibit tumor metabolism, such as carbohydrate and amino acid metabolism, to achieve the clinical goal of inhibiting tumor growth. In particular, for patients with diabetes-associated breast cancer, metabolic disruption of glucose, amino acids, and fat metabolism by diabetes itself can cause breast cancer cells to rely on high glucose and amino acids for growth, and relevant metabolic pathway inhibitors have shown good therapeutic effects."
Below are explanations and quotations of the novel findings included in our review, showing how different inhibitors can inhibit cancer cells proliferation/ kill cancer cells by altering the glucose metabolism.
- Section 1.1: Metformin
- 1st paragraph states that using metformin alone against breast cancer shows insignificant result, reflecting the ineffectiveness of metformin in targeting oncogenes/ tumour suppressor genes
- 2nd and 3rd paragraphs therefore provide novel findings of how metformin can act in combination therapies to target the metabolic reprogramming in breast cancer (Warburg effect), proving that it still has a potential role in DM-associated BC.
- ‘ Glucagon-like peptide-1 receptor agonist exendin-4 (Ex-4), another anti-diabetic drug, has been proven to be effective for BC when used in combination with metformin by inhibiting NF-κB’
- ‘A novel MRS uses metformin and NF‐κB inhibitor to further enhance the rate of aerobic glycolysis and increase the amount of lactate produced. With the addition of an MCT4 inhibitor, lactate accumulation decreases intracellular pH and achieves cytotoxicity by intracellular acidification’
- Section 1.2: GLUT inhibitor
- How we link GLUT inhibitor to diabetes-associated breast cancer
- Metabolism reprogramming in breast cancer cells under diabetic condition:
- ‘Hyperglycemia supports BC cell growth by providing sufficient glucose for aerobic glycolysis, known as the “Warburg effect". Under aerobic glycolysis, a large amount of lactate is produced to generate enough ATP to support rapid cancer cell proliferation’ (Section 1)
- ‘ GLUT1-4 and GLUT12, commonly expressed in BC’
- GLUT1
- ‘upregulated in BC cell lines [1] and plays a significant role in glucose uptake in BC tissues.’
- ‘Confirmed the effectiveness of GLUT1 inhibitors in hindering the growth of triple-negative breast cancer (TNBC)’
- GLUT2
- ‘works well in high glucose concentrations’.
- GLUT2 inhibitors can ‘reduce glucose uptake in BC cell lines like MDA-MB-231[2] and MCF-7 [3].’
- GLUT3
- ‘downregulation of it can inhibit glycolysis and proliferation of BC [4]’
- GLUT4
- ‘GLUT4 inhibition induces metabolic reprogramming, shifts glycolysis to oxidative phosphorylation, and lowers BC's proliferation rate under hypoxic conditions [5]’
- GLUT12
- ‘inhibit TNBC cell proliferation by decreasing glucose uptake and inhibiting aerobic glycolysis [6]’
- GLUT1
- Metabolism reprogramming in breast cancer cells under diabetic condition:
- How we link GLUT inhibitor to diabetes-associated breast cancer
- Section 1.3: SGLT inhibitor
- How we link SGLT inhibitor to DM-associated BC
- Targets the metabolic reprogramming of BC cells where additional glucose is needed for Warburg effect (section 1)
- ‘Hyperglycemia supports BC cell growth by providing sufficient glucose for aerobic glycolysis, known as the “Warburg effect". Under aerobic glycolysis, a large amount of lactate is produced to generate enough ATP to support rapid cancer cell proliferation’
- SGLT2 inhibitors
- ‘inhibit glucose and sodium influx into cells, hyperpolarize cancer cells' membranes and hinder cancer cell growth [7]’
- ‘can inhibit BC proliferation by inducing nutrient deficiency and cell cycle arrest [8]’
- SGLT1 inhibitors
- ‘overexpression of SGLT1 is found in tamoxifen-resistant ER-positive BC. The transporter increases glycolytic flux and lactate production via aerobic glycolysis.'
- ‘Knocking down SGLT1 is proven effective in inhibiting BC growth, including subtypes like TNBC [9] and HER2+ BC [10]’
- Targets the metabolic reprogramming of BC cells where additional glucose is needed for Warburg effect (section 1)
- How we link SGLT inhibitor to DM-associated BC
- Section 1.4: MCT inhibitor
- How we link MCT inhibitor to DM-associated BC
- MCT 1 and 4 are ‘bidirectional lactate transporter’ and ‘lactate exporter’ respectively.
- They transport the lactate produced under ‘the Warburg effect’ where ‘cancer cells shift their metabolism from oxidative phosphorylation to aerobic glycolysis, producing lactate for energy production’
- ‘crucial role in maintaining the balance of lactate and the pH of cells [11].’
- MCT1 inhibitor
- As ‘BC cells rely heavily on aerobic glycolysis and lactate metabolism’, ‘inhibiting lactate import’ by MCT1 inhibitor can ‘reprogram their metabolism to oxidative phosphorylation, which is less favorable for the rapid proliferation rate [12]’ of cancer cells.
- MCT4 inhibitor
- ‘used in combination with metformin and NF-κB inhibitors to achieve intracellular acidification’
- MCT 1 and 4 are ‘bidirectional lactate transporter’ and ‘lactate exporter’ respectively.
- How we link MCT inhibitor to DM-associated BC
- Section 1.5: IGF1R and IR antagonist
- How we link IGF1R and IR antagonist to DM-associated BC treatment
- ‘Hyperinsulinemia in DM promote BC progression [13]’ as ‘insulin could cause tumorigenesis and trigger the downward cascade involving the mTOR signaling pathway’
- IGF1 ‘triggers PI3K/AKT and MAPK signaling pathway, inducing the production of Cyr61’ in which its ‘elevation plays a key role in tumorigenesis.’
- ‘combinations of IGF1R/IR with androgen receptor antagonist or anti-PD-L1, is effective in hindering migration and progression of TNBC cells [14].’
- How we link IGF1R and IR antagonist to DM-associated BC treatment
- 2. “The entire section 2 is also an unnecessary description of amino acids without establishing what link it has with cancer.
Response: The second part of this paper focuses on the impact of various amino acids on tumor metabolism and the potential for targeting their metabolic pathways for cancer treatment in the context of metabolic reprogramming. In addition, we have summarized trials of amino acid pathway treatment by adding an updated Table 2 (Page 12) and Figure 2 (Page 11).
- Quoted below is how we explain the link of amino acids in DM-associated BC
- Newly added in Section 2: ‘Each AA has their specific role in DM-associated BC. Certain AA can provide additional energy to support the rapid cancer cells proliferation. Some of them can also trigger cell signaling pathways which cause malignant cell proliferation.’
- Leucine
- ‘Act as an activator of mTORC1’ which‘lead to insulin resistance and T2DM due to early β-cell apoptosis’. It ‘also increases cell proliferation, growth, and tumorigenesis by various signaling molecules like the eIF4E-binding protein 1.’
- ‘Act as a nutrient source for energy production in cancer cells, augmenting the supply of glucose.’
- Methionine
- ‘An increased level of methionine is often seen in DM. ‘
- Methionine ‘induces tumorigenesis’ ‘via the synthesis of glutathione, formation of polyamine, and donation of a methyl group which promotes DNA methylation’
- ‘Dietary depletion of methionine causes BC cells to rely entirely on the thioredoxin antioxidant pathway instead of the pathway involving glutathione. Combined with a thioredoxin reductase agent, it exerts antitumor effects on BC by increasing oxidative cell stress [15].’
- Cysteine
- ‘Elevated level in DM’
- ‘Cysteine depletion inhibits BC cells' ability to produce glutathione, which protects cells against oxidative stress [16]’ and caused ‘programmed necrosis’.
- Glutamine and glutamate
- ‘In DM, an elevated rate of gluconeogenesis from amino acid is seen under dysregulated glucagon stimulation,’ with glutamine playing ‘an important role’. ‘Increased gluconeogenesis could be a possible mechanism for causing hyperglycemia in DM.’
- In BC cells, ‘glutamine can be converted to ?-ketoglutarate, which incorporates into the TCA cycle to compensate for the insufficient energy production by aerobic glycolysis’
- ‘Reducing glutamine uptake is a means to inhibit TNBC and other BC cell growth.’
- Arginine
- Can be ‘converted to nitric oxide (NO), which plays an important role in tumorigenesis by numerous mechanisms, namely damaging DNA and promoting mutation on tumor suppressor gene, e.g., p53 and angiogenesis.’
- ‘A low NO concentration promotes tumorigenesis, while a high level of it causes cytotoxicity and apoptosis of cancer cells [17].’
- Serine
- One of the ‘key player in biomass synthesis’ which provides additional energy production to support cell growth under the Warburg effect
- Acts as a ‘donor of the one-carbon unit’ to produce nucleotides, synthesize ATP and generate NADH/NADPH which are all essential to support the rapid growth and proliferation of cancer cells
- Asparagine
- A reduced level of asparagine is shown in DM
- ‘Inhibit asparagine synthetase (ASNS) or perform dietary asparagine restriction’ can ‘lower the asparagine level can inhibit the metastasis of BC.’
- Our purpose to discuss disaccording results in the review (e.g. how a treatment may not be effective in patients with both DM and BC) :
- To make it a research gap (stated as ‘more study is needed’). Since there is a limited number of research related to the specific topic, there is a high level of uncertainty.
- Notify people of the current trend (including what may not work) generated from existing but limited evidence related to diabetes-associated breast cancer. This ensures the literature review to be thorough and holistic.
- The authors made no effort to justify the altered metabolic environment in DM and how it contributes to cancer.
Response: We have discussed the altered metabolic environment in DM and how it contributes to cancer.
- How glucose metabolism is altered under DM and BC:
- Stated in Section 1: ‘ Hyperglycemia supports BC cell growth by providing sufficient glucose for aerobic glycolysis, known as the “Warburg effect". Under aerobic glycolysis, a large amount of lactate is produced to generate enough ATP to support rapid cancer cell proliferation [18]. Hyperinsulinemia in DM also promotes BC cell growth. Besides inducing glucose uptake, insulin plays a vital role in activating mTOR via the PI3K/ AKT pathway. Activating mTORC1/2 increases mRNA translation, cellular growth, and cell proliferation and enhances cell survival [19]. Within the pathway, Akt and phospholipase Cγ play a key role in BC patient with diabetic conditions [20].’
- How the amino acid metabolism is altered under DM and BC
- Stated in respective section above regarding how amino acid level changes under diabetic conditions.
- How the lipid metabolism is altered under DM and BC
- Stated in Section 3: ‘Under diabetic alteration of lipid metabolism, there is a ready supply of triglyceride and fatty acid (FA) [21]. FA is an important component for plasma membrane formation and plays a key role in cancer cell growth. It is a fuel for energy production and favors cell survival [13].’
- 4. “The conclusion section lacks clarity”
Response: We greatly appreciate the reviewer’s efforts to carefully review the paper and the valuable suggestions offered. As suggested by the reviewer, we have revised the conclusion section (Page 19).
References
[1] L. Szablewski, Biochimica et Biophysica Acta (BBA)-Reviews on Cancer 2013, 1835 (2), 164.
[2] K.-H. Wu, C.-T. Ho, Z.-F. Chen, L.-C. Chen, J. Whang-Peng, T.-N. Lin, Y.-S. Ho, Journal of food and drug analysis 2018, 26 (1), 221.
[3] C. Azevedo, A. Correia-Branco, J. R. Araújo, J. T. Guimaraes, E. Keating, F. Martel, Nutrition and cancer 2015, 67 (3), 504.
[4] V. Jacquier, D. Gitenay, S. Fritsch, S. Bonnet, B. Győrffy, S. Jalaguier, L. K. Linares, V. Cavaillès, C. Teyssier, Cellular and Molecular Life Sciences 2022, 79 (5), 1.
[5] P. Garrido, F. G. Osorio, J. Morán, E. Cabello, A. Alonso, J. M. Freije, C. González, J Cell Physiol 2015, 230 (1), 191, https://doi.org/10.1002/jcp.24698.
[6] Y. Shi, Y. Zhang, F. Ran, J. Liu, J. Lin, X. Hao, L. Ding, Q. Ye, Cancer Letters 2020, 495, 53, https://doi.org/https://doi.org/10.1016/j.canlet.2020.09.012.
[7] S. Komatsu, T. Nomiyama, T. Numata, T. Kawanami, Y. Hamaguchi, C. Iwaya, T. Horikawa, Y. Fujimura-Tanaka, N. Hamanoue, R. Motonaga, M. Tanabe, R. Inoue, T. Yanase, D. Kawanami, Endocrine Journal 2020, 67 (1), 99, https://doi.org/10.1507/endocrj.EJ19-0428.
[8] J. Zhou, J. Zhu, S. J. Yu, H. L. Ma, J. Chen, X. F. Ding, G. Chen, Y. Liang, Q. Zhang, Biomed Pharmacother 2020, 132, 110821, https://doi.org/10.1016/j.biopha.2020.110821.
[9] H. Liu, A. Ertay, P. Peng, J. Li, D. Liu, H. Xiong, Y. Zou, H. Qiu, D. Hancock, X. Yuan, W. C. Huang, R. M. Ewing, J. Downward, Y. Wang, Mol Oncol 2019, 13 (9), 1874, https://doi.org/10.1002/1878-0261.12530.
[10] J. Wang, H. Ji, X. Niu, L. Yin, Y. Wang, Y. Gu, D. Li, H. Zhang, M. Lu, F. Zhang, Q. Zhang, Dis Markers 2020, 2020, 6103542, https://doi.org/10.1155/2020/6103542.
[11] A. P. Halestrap, M. C. Wilson, IUBMB Life 2012, 64 (2), 109, https://doi.org/10.1002/iub.572.
[12] J. Padilla, B. S. Lee, K. Zhai, B. Rentz, T. Bobo, N. M. Dowling, J. Lee, Cells 2022, 11 (7), https://doi.org/10.3390/cells11071177.
[13] S. D. Martin, S. L. McGee, Journal of Endocrinology 2018, 237 (2), R35.
[14] a) N. M.-G. Hamilton, Diana; Rogers, Ben; Austin, David; Foos, Kay; Tong, Ashley; Adams, Diana; Vadgama, Jaydutt; Brecht, Mary-Lynn; Pietras, Richard, SPG BioMed 2019; b) N. M. Hamilton, D. C. Marquez-Garban, L. P. Burton, B. Comin-Anduix, A. J. Garcia, J. V. Vadgama, Cancer Research 2020, 80 (16_Supplement), LB, https://doi.org/10.1158/1538-7445.AM2020-LB-391.
[15] D. Malin, Y. Lee, O. Chepikova, E. Strekalova, A. Carlson, V. L. Cryns, Breast Cancer Res Treat 2021, 190 (3), 373, https://doi.org/10.1007/s10549-021-06398-y.
[16] X. Tang, C. K. Ding, J. Wu, J. Sjol, S. Wardell, I. Spasojevic, D. George, D. P. McDonnell, D. S. Hsu, J. T. Chang, J. T. Chi, Oncogene 2017, 36 (30), 4235, https://doi.org/10.1038/onc.2016.394.
[17] F. H. Khan, E. Dervan, D. D. Bhattacharyya, J. D. McAuliffe, K. M. Miranda, S. A. Glynn, Int J Mol Sci 2020, 21 (24), https://doi.org/10.3390/ijms21249393.
[18] O. Warburg, Science 1956, 123 (3191), 309.
[19] J. Dancey, Nature reviews Clinical oncology 2010, 7 (4), 209.
[20] N. M. Tomas, K. Masur, J. C. Piecha, B. Niggemann, K. S. Zänker, BMC research notes 2012, 5 (1), 1.
[21] K. G. Parhofer, Diabetes Metab J 2015, 39 (5), 353, https://doi.org/10.4093/dmj.2015.39.5.353.

Round 2
Reviewer 2 Report
Maybe accepted after incorporating the following information
1. Given the fact that altered metabolism could affect immune network , include a short description on immune dysregulation in DM and how that maybe associated with cancer progression (involving the immune system).
Few References :
PMID: 34885023
PMID: 24655299
PMID: 35957877
https://doi.org/10.1186/s12943-021-01316-8
Author Response
Response: We greatly appreciate the reviewer's insightful comments. As suggested by the reviewer, we have added the following paragraph in our revised manuscript [marked] (Page 19).
Immune dysregulation in diabetes that maybe associated with cancer progression
In the presence of tumor, anti-tumor immune cells mediate inflammatory responses to kill cancer cells after activating a metabolic switch. Nevertheless, tumors will develop strategies to avoid such damage. Cancer cells can change the metabolic environment of tumors through sequestering nutrients (such as glucose, tryptophan, arginine) and producing toxic wastes (such as adenosine, lactic acid, kynurenine). This tumor environment promotes the depletion of anti-tumor immune cells, the expansion of Tregs and the expression of immune checkpoints [1-3]. The establishment of this immunosuppressive tumor environment reduces the therapeutic response of cancer patients to immunotherapy. The metabolic disorder under the condition of diabetes coordinates and aggravates this process [4]. In addition, diabetes mellitus itself can directly lead to immune dysregulation. For example, high glucose conditions in diabetes affect protein-oligosaccharide interactions via competitive inhibition [5]. Moreover, the increase of blood glucose level will form covalent sugar adducts with some proteins via nonenzymatic glycation. This can damage humoral immunity in various ways, for example, by changing the structure and function of immunoglobulins [6-9]. Previous studies have revealed that T cell function in patients with T2DM is impaired [10-12]. More and more evidence shows that abnormal metabolism in diabetes not only plays a key role in maintaining cancer signals of tumor occurrence and survival, but also has a wider significance in regulating anti-tumor immune response by releasing metabolites and influencing the expression of immune molecules, for instance lactic acid, PGE2 and arginine [13]. Therefore, combination therapy between immunotherapy and metabolic intervention can be employed to better release the potential of anticancer therapy. Knowledge of immunometabolism allows novel therapeutic strategies to improve anti-tumor immune response through targeting the metabolism of tumor and immune cells, so as to improve immunotherapy.
References
- Luby A, Alves-Guerra MC: Targeting Metabolism to Control Immune Responses in Cancer and Improve Checkpoint Blockade Immunotherapy. Cancers (Basel) 2021, 13.
- Ganeshan K, Chawla A: Metabolic regulation of immune responses. Annu Rev Immunol 2014, 32:609-634.
- Sahoo OS, Pethusamy K, Srivastava TP, Talukdar J, Alqahtani MS, Abbas M, Dhar R, Karmakar S: The metabolic addiction of cancer stem cells. Front Oncol 2022, 12:955892.
- Wu Y, Dong Y, Atefi M, Liu Y, Elshimali Y, Vadgama JV: Lactate, a Neglected Factor for Diabetes and Cancer Interaction. Mediators Inflamm 2016, 2016:6456018.
- Ilyas R, Wallis R, Soilleux EJ, Townsend P, Zehnder D, Tan BK, Sim RB, Lehnert H, Randeva HS, Mitchell DA: High glucose disrupts oligosaccharide recognition function via competitive inhibition: a potential mechanism for immune dysregulation in diabetes mellitus. Immunobiology 2011, 216:126-131.
- Daryabor G, Atashzar MR, Kabelitz D, Meri S, Kalantar K: The Effects of Type 2 Diabetes Mellitus on Organ Metabolism and the Immune System. Front Immunol 2020, 11:1582.
- Shcheglova T, Makker S, Tramontano A: Reactive immunization suppresses advanced glycation and mitigates diabetic nephropathy. J Am Soc Nephrol 2009, 20:1012-1019.
- Vrdoljak A, Trescec A, Benko B, Hecimovic D, Simic M: In vitro glycation of human immunoglobulin G. Clin Chim Acta 2004, 345:105-111.
- Lapolla A, Tonani R, Fedele D, Garbeglio M, Senesi A, Seraglia R, Favretto D, Traldi P: Non-enzymatic glycation of IgG: an in vivo study. Horm Metab Res 2002, 34:260-264.
- Kumar NP, Sridhar R, Nair D, Banurekha VV, Nutman TB, Babu S: Type 2 diabetes mellitus is associated with altered CD8(+) T and natural killer cell function in pulmonary tuberculosis. Immunology 2015, 144:677-686.
- Moura J, Rodrigues J, Goncalves M, Amaral C, Lima M, Carvalho E: Impaired T-cell differentiation in diabetic foot ulceration. Cell Mol Immunol 2017, 14:758-769.
- Richard C, Wadowski M, Goruk S, Cameron L, Sharma AM, Field CJ: Individuals with obesity and type 2 diabetes have additional immune dysfunction compared with obese individuals who are metabolically healthy. BMJ Open Diabetes Res Care 2017, 5:e000379.
- Xia L, Oyang L, Lin J, Tan S, Han Y, Wu N, Yi P, Tang L, Pan Q, Rao S, et al: The cancer metabolic reprogramming and immune response. Mol Cancer 2021, 20:28.